# Folding and Binding Mechanisms of the SH2 Domain from Crkl

**DOI:** 10.3390/biom12081014

**Published:** 2022-07-22

**Authors:** Caterina Nardella, Angelo Toto, Daniele Santorelli, Livia Pagano, Awa Diop, Valeria Pennacchietti, Paola Pietrangeli, Lucia Marcocci, Francesca Malagrinò, Stefano Gianni

**Affiliations:** Istituto Pasteur—Fondazione Cenci Bolognetti, Dipartimento di Scienze Biochimiche “A. Rossi Fanelli” and Istituto di Biologia e Patologia Molecolari del CNR, Sapienza Università di Roma, Piazzale A. Moro 5, 00185 Rome, Italy; caterina.nardella@uniroma1.it (C.N.); angelo.toto@uniroma1.it (A.T.); daniele.santorelli@uniroma1.it (D.S.); livia.pagano@uniroma1.it (L.P.); awa.diop@uniroma1.it (A.D.); valeria.pennacchietti@uniroma1.it (V.P.); paola.pietrangeli@uniroma1.it (P.P.); lucia.marcocci@uniroma1.it (L.M.)

**Keywords:** kinetics, fluorescence, site-directed mutagenesis, protein–protein interactions, SH2 domains, Crkl, Paxillin

## Abstract

SH2 domains are structural modules specialized in the recognition and binding of target sequences containing a phosphorylated tyrosine residue. They are mostly incorporated in the 3D structure of scaffolding proteins that represent fundamental regulators of several signaling pathways. Among those, Crkl plays key roles in cell physiology by mediating signals from a wide range of stimuli, and its overexpression is associated with several types of cancers. In myeloid cells expressing the oncogene BCR/ABL, one interactor of Crkl-SH2 is the focal adhesion protein Paxillin, and this interaction is crucial in leukemic transformation. In this work, we analyze both the folding pathway of Crkl-SH2 and its binding reaction with a peptide mimicking Paxillin, under different ionic strength and pH conditions, by using means of fluorescence spectroscopy. From a folding perspective, we demonstrate the presence of an intermediate along the reaction. Moreover, we underline the importance of the electrostatic interactions in the early event of recognition, occurring between the phosphorylated tyrosine of the Paxillin peptide and the charge residues of Crkl-SH2. Finally, we highlight a pivotal role of a highly conserved histidine residue in the stabilization of the binding complex. The experimental results are discussed in light of previous works on other SH2 domains.

## 1. Introduction

The metabolism of cells is primarily regulated by the specific recognition of their constituents. To achieve this task, proteins often display protein–protein recognition domains, which are critical in the assembly of numerous intracellular complexes that mediate diverse cellular processes [1]. Among others, SH2 domains represent an abundant class of protein–protein recognition domains, consisting of about 100 amino acids, known to bind characteristic protein motifs containing phosphorylated tyrosine residues [2,3]. Dysregulated interactions mediated by SH2 domains have been associated with several human diseases [4,5,6], posing this class of protein domain as a very interesting target for drug discovery [7,8,9].

Crkl is a ubiquitously expressed adaptor protein belonging to the proto-oncogene Crk family, composed of one SH2 domain at the N-terminus followed by two SH3 domains, N-SH3 and C-SH3, that are specialized to recognize proline-rich protein motifs [10]. Whilst Crkl does not possess any enzymatic activity, its physiological role is nevertheless very important, as it participates in several signal transduction networks by binding specific protein ligands and acting as their spatial and temporal regulator [11]. Thus, it is well-established that, together with its interaction partners, Crkl plays a key role in several processes, such as cell proliferation, apoptosis, cell adhesion and migration [12,13].

Being a so-called “adaptor protein”, Crkl exerts its functions primarily via its protein–protein interaction domains. Of interest, the SH2 domain of Crkl (Crkl-SH2) is an important mediator of the phosphorylated tyrosine-dependent signaling [14]. From a structural perspective, the SH2 domain of Crkl is characterized by the conserved fold of its superfamily, consisting of a central 4/6-stranded antiparallel β-sheet flanked by two α helices, with characteristic loops joining the secondary structural elements [15,16,17]. Previous studies have already identified different interaction partners of Crkl-SH2 that exert a relevant physiological role [12,13,18,19]. Nevertheless, no detailed description of the interaction mechanisms has been provided to date.

Among others, a particularly interesting partner of Crkl-SH2 is represented by the BCR/ABL oncoprotein, which is dysfunctional in chronic myelogenous leukemia [20]. In fact, Crkl-SH2 is critical in linking BCR/ABL to the focal adhesion protein Paxillin, contributing to leukemic transformation. On the other hand, Paxillin itself recruits other proteins to the focal adhesions, such as cytoskeletal proteins, tyrosine/serine/threonine kinases, GTPase activating proteins and others, thus playing a pivotal role in several signaling pathways [21].

In this work, we provide a complete characterization of Crkl-SH2 from both a folding and a functional perspective. In particular, we address the mechanism of folding of this domain under different experimental conditions, as well as its binding reaction with a peptide mimicking Paxillin. Furthermore, by taking advantage of site-directed mutagenesis, we highlight the critical role of His60 Crkl-SH2 in the binding reaction. Such amino acid is conserved within the SH2 domain family [2,22]. Our unfolding and binding data are discussed in light of previous works on other SH2 domains.

## 2. Materials and Methods

### 2.1. Protein Expression and Purification

The construct encoding the SH2 domain of Crkl protein (residues 1−111) was subcloned in a pET28b+ plasmid vector and then transformed in *Escherichia coli* cells BL21 (DE3). Bacterial cells were grown in LB medium containing 30 μg/mL of kanamycin at 37 °C until OD600 = 0.7−0.8, and protein expression was then induced with 0.5 mM IPTG. After induction, cells were grown at 25 °C overnight and then collected by centrifugation. To purify the His-tagged protein, the pellet was resuspended in buffer made of 50 mM TrisHCl, 300 mM NaCl and 10 mM Imidazole, pH 7.5, with the addition of antiprotease tablet (Complete EDTA-free, Roche), then sonicated and centrifuged. The soluble fraction from bacterial lysate was loaded onto a nickel-charged HisTrap Chelating HP (GE Healthcare) column equilibrated with 50 mM TrisHCl, 300 mM NaCl and 10 mM Imidazole, pH 7.5. Protein was then eluted with a gradient from 0 to 1 M imidazole using an ÄKTA-prime system. Fractions containing the protein were collected, and the buffer was exchanged to 50 mM TrisHCl, 300 mM NaCl, pH 7.5, using a HiTrap Desalting column (GE Healthcare). The purity of the protein was analyzed through SDS-page. Site-directed mutagenesis was performed using the QuikChange mutagenesis kit (Agilent Technologies Inc., Santa Clara, CA, USA), according to the manufacturer’s instructions. To increase protein solubility, all the experiments were carried out on the N-terminal covalently bound his-tagged protein. Peptides mimicking the region 112–123 of Paxillin, with and without the dansyl N-terminal modification, were purchased from GenScript.

### 2.2. Equilibrium Unfolding Experiments

Equilibrium unfolding experiments were performed on a Fluoromax single photon counting spectrofluorometer (Jobin-Yvon, Edison, NJ, USA). The SH2 domain was excited at 280 nm, and emission spectra were recorded between 300 and 400 nm at increasing guanidine-HCl concentrations. Experiments were performed with the protein at a constant concentration of 2 μM, at 298 K, using a quartz cuvette with a path length of 1 cm. Buffers containing 0.15 M sodium-sulphate used for pH dependence were: 50 mM sodium-acetate pH 4.0, 50 mM sodium-acetate pH 4.5, 50 mM sodium-acetate pH 5.0, 50 mM sodium-acetate pH 5.5, 50 mM sodium-phosphate pH 6.7, 50 mM sodium-HEPES pH 7.2, 50 mM TrisHCl pH 8.0, 50 mM TrisHCl pH 8.5, 50 mM TrisHCl pH 9.0. Data were fitted using Equation (1):(1)$$Y_{obs}=\frac{(Y_{N}+\alpha_{N}\left[GdnHCl\right])+(Y_{D}+\alpha_{D}\left[GdnHCl\right])\;e^{\frac{m_{D-N}\left(\left[GdnHCl\right]-{\left[GdnHCl\right]}_{1/2}\right)}{RT}}}{\begin{array}{c} 1+e^{\frac{m_{D-N}\left(\left[GdnHCl\right]-{\left[GdnHCl\right]}_{\frac{1}{2}}\right)}{RT}} \end{array}}$$
where: *Y_obs_* is the observed fluorescence signal; *Y_N_* and *Y_D_* are the fluorescence signals of the native and denatured states, respectively; $\alpha_{N}=\frac{\partial \;Y_{N}}{\partial \;\left[GdnHCl\right]}$ and $\alpha_{D}=\frac{\partial \;Y_{D}}{\partial \;\left[GdnHCl\right]}$
**;** [*GdnHCl*]_1/2_ is the denaturant concentration at which the protein is 50% unfolded. The equation may be derived by assuming the presence of a two-state mechanism, with the change in free energy between the two states varying linearly with denaturant concentration, with a slope of *m_D-N_* kcal mol^−1^ M^−1^.

### 2.3. Stopped-Flow (Un)Folding Experiments

Kinetic (un)folding experiments were performed on an Applied Photophysics Pi-star 180 stopped-flow apparatus, monitoring the change of fluorescence emission, exciting the sample at 280 nm, and recording the fluorescence emission using a 360 (for acidic conditions: 4.0, 4.5, 5.0 pH) or 320 nm (for other pH conditions) cutoff glass filter. The experiments were performed at 298 K using guanidine-HCl as denaturant agent. The buffers used were the same described in the Equilibrium unfolding experiment paragraph. Data were collected in the absence and in the presence of 0.15 M sodium-sulphate. For each denaturant concentration, at least five individual traces were averaged. The final protein concentration was typically 3 μM. Data were fitted using Equation (2):(2)$$K_{obs}=\frac{k_{IN}\;e^{\frac{-m_{I-N}\left(\left[GdnHCl\right]\right)}{RT}}}{1+K_{\begin{array}{c} IU \end{array}}\;e^{\frac{m_{I-D}\left(\left[GdnHCl\right]\right)}{RT}}}+k_{NI}\;e^{\frac{m_{N-I}\left(\left[GdnHCl\right]\right)}{RT}\;}$$
where *K_IU_* = *k_IU_*/*k_UI_*, with *k_IU_* and *k_UI_* respectively representing the folding and unfolding rate constants from the denatured state to the intermediate state.

### 2.4. Stopped-Flow Binding Experiments

Kinetic binding experiments were performed on an Applied Photophysics sequential-mixing DX-17MV stopped-flow apparatus (Applied Photophysics, Leatherhead, UK), set up in single mixing mode. Pseudo-first-order binding experiments were performed mixing a constant concentration (2 μM) of dansyl-Pax_112–123_ with increasing Crkl-SH2, from 2 to 10 μM. Samples were excited at 280 nm, and the emission fluorescence was recorded using a 475 nm cutoff filter, recording fluorescence above 475 nm. Experiments were performed at 283 K. The buffers containing 0.5 M NaCl used for pH dependence were: 50 mM sodium-acetate pH 5.0, 50 mM sodium-acetate pH 5.5, 50 mM BisTRIS pH 6.0, 50 mM BisTRIS pH 6.8, 50 mM sodium-HEPES pH 7.5, 50 mM TrisHCl pH 8.0, 50 mM TrisHCl pH 8.5, 50 mM TrisHCl pH 9.0. For ionic strength dependence, buffers used were 50 mM sodium-HEPES pH 7.5 150 mM NaCl, 50 mM sodium-HEPES pH 7.5 300 mM NaCl, 50 mM sodium-HEPES pH 7.5 500 mM NaCl and 50 mM sodium-HEPES pH 7.5 1 M NaCl. For each acquisition, five traces were collected, averaged, and satisfactorily fitted to a single-exponential equation with flat residuals and typically displaying an R > 0.98.

### 2.5. Stopped-Flow Displacement Experiments

As mentioned in the text, microscopic dissociation rate constants (*k*_off_) were measured by performing displacement experiments on an Applied Photophysics sequential-mixing DX-17MV stopped-flow apparatus (Applied Photophysics), set up in single mixing mode. A preincubated complex of SH2 domain and dansyl-Pax_112–123_ at a constant concentration of 2 µM was rapidly mixed with an excess of non-dansylated Pax_112–123_, (30 μM). For displacement experiments concerning the H60A SH2 mutant, the binding complex was formed by incubating 2 µM mutant protein and 20 µM dansyl-Pax_112–123_. Then, the resulting complex was rapidly mixed with an excess of non-dansylated Pax_112–123_ (40 μM). Samples were excited at 280 nm, and fluorescence emission was collected using a 475 nm cutoff filter. Experiments were performed at 283 K. The observed rate constants were calculated from the average of five single traces. Observed kinetics were consistent with a single-exponential decay with flat residuals and typically displaying an R > 0.98. Folding and unfolding experiments did not show any dependence of the observed rate constant when performed at different protein concentrations, indicating a lack of transient aggregation events.

## 3. Results

### 3.1. The Folding Pathway of Crkl SH2 Domain

We investigated the mechanism of folding of Crkl-SH2 using both equilibrium and kinetic experiments. Given that the SH2 domain contains a tryptophan residue at the 14 position, the unfolding and folding processes were spectroscopically followed by monitoring the intrinsic fluorescence of this residue upon excitation at 280 nm. In the equilibrium unfolding experiments, the emission fluorescence of Crkl-SH2 (2 μM) was recorded in the range 300–400 nm at increasing denaturant concentrations (from 0 to 5.6 M Gnd-HCl) and 298 K. Then, denaturation curves for Crkl-SH2 were obtained by plotting the emission fluorescence at five different wavelengths (ranging from 320 to 360 nm) versus the guanidine-HCl concentrations (Figure 1A). Observed data could be satisfactorily fitted to a sigmoidal equation by sharing the midpoint and m_D-N_ value, suggesting that the equilibrium unfolding of Crkl-SH2 is consistent with a two-state reaction, without any detectable intermediates significantly accumulating. In particular, curve fitting returned flat residuals with a value of R typically higher than 0.97.

To infer the robustness of the equilibrium unfolding of Crkl-SH2, we monitored the denaturation under different pH conditions and extrapolated the relative m_D-N_ values and midpoints by fitting the unfolding curves with the two-state model. Whilst it is evident that pH modulates the stability of Crkl-SH2, as shown by the change in the denaturation midpoint, the m_D-N_ values were relatively robust to changes in experimental conditions, reinforcing the two-state nature of the equilibrium transition (Figure 1B). The m_D-N_ value calculated from global fitting at different pH values was 3.4 ± 0.1 Kcal mol^−1^ M^−1^, which is consistent with a protein of 111 residues [23].

To address the mechanism of folding of Crkl-SH2, we carried out time-resolved fluorescence monitored stopped-flow experiments at different ionic strengths and pH conditions. Under all the investigated conditions, the refolding and unfolding traces could be satisfactorily fitted to a single-exponential process. Figure 2 and Figure 3 report the logarithms of observed rate constants plotted against the Gdn-HCl concentration to generate chevron plots. Interestingly, a clear deviation from linearity, classically denoted as a “roll-over effect” [24,25], was evident in the refolding branch at a low concentration of the denaturing agent. This finding represents a classical signature for the presence of a partially folded intermediate whose accumulation transiently occurs at a specific denaturant concentration and parallels what was previously observed in the case of the folding mechanism of the N-SH2 domain from SHP2 [26]. Inspection of Figure 2 shows that, as expected, the addition of sodium-sulphate determines a stabilization of both the native and intermediate states and allows an inference of the roll-over effect with a remarkably increased level of confidence. Hence, we carried out complete pH dependence both in the presence (Figure 3A) and in the absence (Figure 3B) of this stabilizing salt. All data were fitted to a kinetic three-state model as formalized in Equation (2), and we calculated the kinetic parameters referring to the rate constants *k*_IN_ and *k*_NI_, the equilibrium constant K_IU_ (K_IU_ = *k*_IU_/*k*_UI,_) and their associated *m* values (Appendix A).

### 3.2. The Binding Reaction between the SH2 Domain of Crkl and Paxillin

To elucidate the details of the interaction occurring between the SH2 domain of Crkl and Paxillin, we monitored binding with a peptide mimicking a specific region of Paxillin, ranging from residues 112 to 123 and carrying a dansyl fluorophore covalently attached to the N-terminus (Pax_112–123_ N_TERM_-Dans-GEEEHV-pY-SFPNK-C_TERM_). In analogy to what was described in our previous works [26,27], the binding kinetics were followed spectroscopically with the stopped-flow apparatus by measuring the change in the FRET (Fluorescence Resonance Energy Transfer) signal upon binding. In this system, the energy is transferred from the tryptophan residue in position 14 of the SH2 domain (donor) to the dansyl group of the Pax_112–123_ peptide (acceptor) when binding occurs. By following this approach, a fixed concentration of dansylated Pax_112–123_ (2 µM) was rapidly mixed with increasing concentrations of the SH2 domain (from 2 to 10 µM) at 283 K. To test the contribution of electrostatic interactions in the complex formation, binding kinetic experiments were carried out at different ionic strengths (0.15, 0.3, 0.5 and 1 M sodium-chloride) and pH (from 5 to 9 pH) conditions. All the binding traces obtained by the time-resolved fluorescence monitoring were satisfactorily fitted with a single-exponential equation with flat residuals and typically displaying an R > 0.98, which allowed the extrapolation of the observed rate constants (*k*_obs_). Then, the *k*_obs_ values were plotted versus different concentrations of the SH2 domain (Figure 4), and data were fitted by the following linear function:$$k_{obs}=k_{\mathrm{on}}\;\left[C-SH2\right]+k_{\mathrm{off}}$$
with the slope and the *y*-axis intercept of the line representing the microscopic association (*k*_on_) and dissociation rate constants (*k*_off_), respectively. Given the high experimental error associated with indirect measurements of the dissociation rate constants, the *k*_off_ values were determined through displacement experiments by mixing a pre-formed Crkl-SH2 domain/dansylated Pax_112–123_ complex with a high excess of non-dansylated Pax_112–123_ peptide, as detailed in the Materials and Methods section. Our data displayed a clear decrease in the microscopic association rate constant as the ionic strength increased, while the value of *k*_off_ was only marginally affected (Figure 4A, Appendix A). In agreement with what has already been reported in the literature for the binding reaction of other SH2 domains [26,27], these results confirm the importance of electrostatic interactions in complex formation.

The dependence of the logarithm of *k*_on_ and *k*_off_ versus pH is reported in Figure 5A (wt). Fitting the curves with the Henderson–Hasselbalch equation returns a pK_a_ of 7.1 ± 0.2, a value close to the pK_a_ of the side chain of histidine (6.04). Given that the Crkl-SH2 domain contains four histidine residues (His33, His60, His80 and His91) (Figure 5B), these were individually replaced by alanine residues through site-directed mutagenesis with the aim of testing their role in the binding reaction. Thus, four variants of the Crkl-SH2 domain were produced (H33A, H60A, H80A and H91A) and used in kinetic binding experiments to monitor the effect of the mutation under different pH conditions. The *k*_on_ and *k*_off_ values obtained by the binding reactions of the H33A, H80A and H91A variants are listed in Table 1. The dependence of these values as a function of pH showed a sigmoidal behavior similar to that observed for the wild-type Crkl SH2 domain (Figure 5A). All curves were satisfactorily fitted with the same equation by sharing the pK_a_ value of 7.1. Notably, the observed pKa is very different from the N- and C-termini of the peptide, indicating that the observed dependence is not affected by the presence of such electrostatic dipole.

On the contrary, for the binding reaction of the H60A mutant with dansylated Pax_112–123_, no binding traces could be recorded with the stopped-flow apparatus. However, the formation of the complex between the H60A mutant and Pax_112–123_ was revealed through displacement experiments (as explained in Materials and Methods) carried out under three different pH conditions, 5.5, 6.8 and 7.5. The resulting dissociation rate constant values (212 ± 11, 152 ± 10, 237 ± 21 s^−1^, respectively) were approximately eight-fold higher than those obtained with the wild-type form of Crkl-SH2 under the same experimental conditions. Furthermore, in the case of the H60A variant, the dissociation rate constants appeared essentially insensitive to pH, suggesting that the pH dependence observed in the case of the wild-type Crkl-SH2 and the H33A, H80A and H91A variants may be ascribed to the protonation of His60. This finding is consistent with the proximity of His60 to the ligand, which may be observed in the complex depicted in Figure 5B.

## 4. Discussion

A powerful strategy for unveiling the folding and function of protein domains is represented by the comparison of homologous proteins sharing the same topology and showing different primary structures. From a folding perspective, it appears that the mechanism of folding of Crkl-SH2 is reminiscent to what was previously observed in the case of the N-SH2 of SHP2 [26], with both proteins displaying a pronounced roll-over effect in the refolding branch of the chevron plot. This effect, which is clearly more visible under stabilizing conditions, can nevertheless be detected under different experimental conditions, indicating that intermediate formation is rather robust to changes in protein stability. It should be noticed, however, that previous folding analysis on other SH2 domains have shown that intermediate formation may not be mandatory for the folding of this class of proteins, as exemplified in the case of the SH2 domains from Src and p85 [28,29], which both fold via a two-state mechanism. Notably, the apparent similarity between the folding mechanisms of Crkl-SH2 with N-SH2 of SHP2 does not correspond to a relevant sequence identity, with these two proteins sharing only 24.7% sequence identity. On the other hand, such value is remarkably lower than the percent sequence identity between the N-terminal and C-terminal SH2 domain of SHP2 (41.2%), whose chevron plots has appeared remarkably different among them and displayed an unfolding roll-over in the case of C-SH2 and a refolding roll-over in the case of N-SH2. This finding reinforces the notion that small changes in sequence composition may have profound effects in folding intermediate stability [30].

Because of the importance of Crkl in orchestrating several metabolic pathways [11,12,13], it is critical to quantitatively establish its interactions with relevant physiological partners. In this context, the results presented in this work allowed a depiction of the mechanism of interaction between Crkl-SH2 and Paxillin, as well as a comparison of them with previously characterized SH2 domains. The kinetic binding experiments indicated that the microscopic association rate constant (*k*_on_) was substantially reduced as the ionic strength increased, whilst the microscopic dissociation rate constant (*k*_off_) was not affected by the sodium-chloride concentrations. These observations allow the conclusion that the transition state of the binding reaction is primarily stabilized by the electrostatic recognition between the interacting partners, possibly between the phospho-tyrosine of the peptide and the charge residues of the SH2 domain, contained in its binding pocket. Furthermore, the analysis of our kinetic data obtained under different pH conditions showed that the protonation state of a histidine residue is critical in balancing both the *k*_on_ and *k*_off_ rate constants of the binding reaction. In this respect, mutational analysis of the domain suggests that such an effect should be ascribed to the protonation of His60, which is located at the binding site of Crkl-SH2. These findings may be important in elucidating the fine details that determine the stability of Crkl-SH2 for its ligand and, therefore, may contribute to the design of potential inhibitors in these regions.

To conclude, the results presented in this work, together with the analysis of previous results, allowed a depiction of the general features of the folding pathway of SH2 domains. Furthermore, we contributed a quantitative description of a critical interaction that is pivotal for the activation of the dysfunctional signaling pathways mediated by BCR/ABL. In this context, we pave the way for future works aimed at designing inhibitors of this aberrant interaction.

## Figures and Tables

**Figure 1 biomolecules-12-01014-f001:**
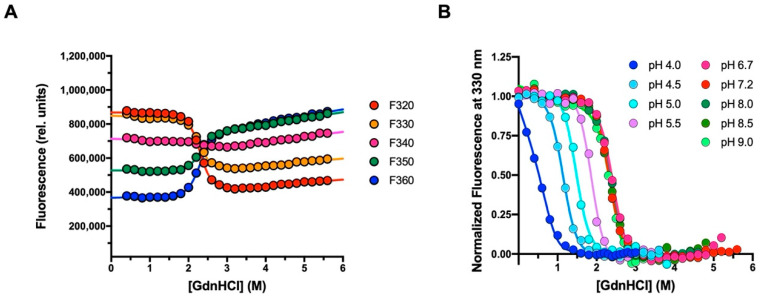
Equilibrium unfolding experiments of the Crkl SH2 domain carried out at 298 K, using guanidine-HCl as denaturing agent. (**A**) Equilibrium denaturation performed in buffer 50 mM sodium-HEPES pH 7.2 containing 0.15 M Na_2_SO_4_. Different emission wavelengths are plotted against the concentration of guanidine-HCl. The unfolding curves were fitted with a two-state model equation (Equation (1)), sharing the m_D-N_ value and midpoint for all datasets. (**B**) Equilibrium denaturation curves collected at different pH conditions in presence of 0.15 M Na_2_SO_4_. The normalized fluorescence recorded at 330 nm is shown as a function of guanidine-HCl concentrations. Lines represent the best fit to a two-state transition (Equation (1)) by sharing the m_D-N_ value between all datasets. The global m_D-N_ value calculated is 3.4 ± 0.1 Kcal mol^−1^ M^−1^.

**Figure 2 biomolecules-12-01014-f002:**
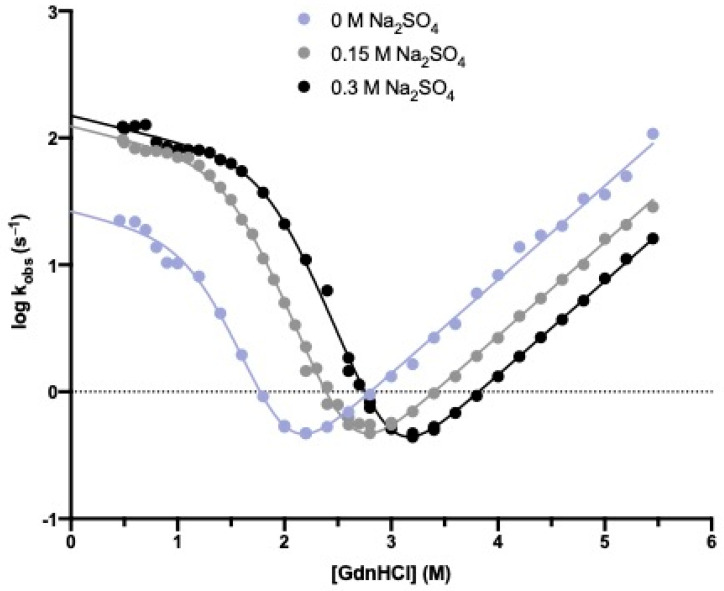
Kinetic (un)folding experiments of the Crkl SH2 domain carried out at 298 K in buffer 50 mM sodium-HEPES pH 7.2 containing different sodium-sulphate concentrations. The logarithm of the observed rate constants measured with the stopped-flow apparatus is plotted versus the concentration of guanidine-HCl. The lines are the best fit to a three-state model as formalized in Equation (2). The related kinetic parameters are listed in Appendix A. For each acquisition, five traces were collected, averaged and satisfactorily fitted to a single-exponential equation.

**Figure 3 biomolecules-12-01014-f003:**
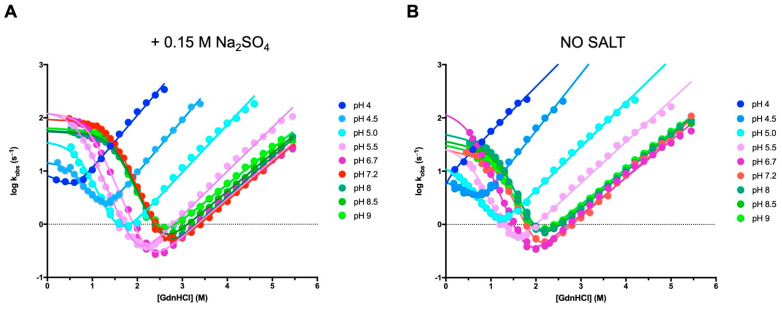
Kinetic (un)folding experiments of the Crkl SH2 domain at different pH conditions and 298 K. The logarithm of the observed rate constants measured with the stopped-flow apparatus is plotted versus the concentration of guanidine-HCl, in the presence (**A**) and absence (**B**) of 0.15 M Na_2_SO_4_. The lines are the best fit to a three-state model as formalized in Equation (2). The kinetic parameters referring to chevron plots shown in (**A**) are listed in Appendix A. For each acquisition, five traces were collected, averaged and satisfactorily fitted to a single-exponential equation.

**Figure 4 biomolecules-12-01014-f004:**
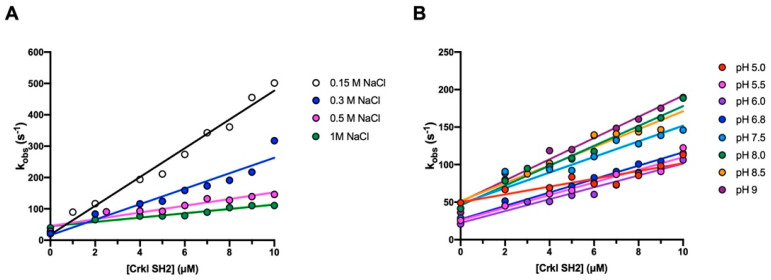
Kinetic binding experiments between the Crkl-SH2 domain and Pax_112–123_ peptide. The pseudo-first-order reactions were measured by mixing 2 μM of Dans Pax_112–123_ with increasing protein concentrations (ranging from 2 to 10 μM), at different ionic strengths (**A**) and pH (**B**) conditions. Lines represent the best fit to a linear equation. The related kinetic parameters are listed in Appendix A and Table 1, respectively. For each acquisition, five traces were collected, averaged, and satisfactorily fitted to a single-exponential equation.

**Figure 5 biomolecules-12-01014-f005:**
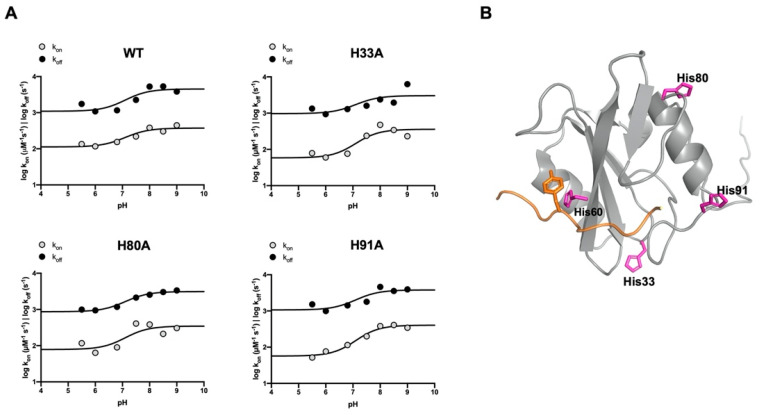
The role of histidine residues of Crkl-SH2 in the binding reaction with Paxillin. (**A**) The pH dependence of the binding reaction of wild-type and His-to-Ala SH2 domain mutants with Pax_112–123_ peptide. The logarithm of microscopic association (gray circles) and dissociation (black circles) rate constants is reported as a function of pH. Curves are the best fit to the Henderson–Hasselbalch equation. The related kinetic parameters are listed in Table 1. (**B**) The three-dimensional structure of the Crkl SH2 domain (PDB code: 2EO3) was superimposed to that of the SHP2 N-SH2 domain in complex with GAB1 peptide (not shown; PDB code: 4QSY) to highlight the conserved binding pocket in the Crkl SH2 domain. In orange, a general ligand containing a phospho-tyrosine residue is shown. All the histidine residues of the Crkl SH2 domain (His 33, His 60, His 80 and His 91) are represented as sticks colored magenta.

**Table 1 biomolecules-12-01014-t001:** Kinetics parameters obtained from pseudo-first-order binding reaction of the wild-type Crkl SH2 domain and histidine-to-alanine mutants with Pax_112–123_ peptide, at different pHs and 283 K.

WT				H33A			
pH	*k*_on_ (μM^−1^ s^−1^)	*k*_off_ (s^−1^)	K_D_ (μM)	pH	*k*_on_ (μM^−1^ s^−1^)	*k*_off_ (s^−1^)	K_D_ (μM)
5.0	5.2 ± 0.9	49.1 ± 1.7	9.5 ± 1.7	5.0	4.4 ± 0.3	33.2 ± 1.4	7.6 ± 0.7
5.5	8.4 ± 0.7	25.6 ± 0.4	3.0 ± 0.3	5.5	6.7 ± 0.4	22.8 ± 2.2	3.4 ± 0.4
6.0	7.9 ± 0.6	20.8 ± 0.3	2.6 ± 0.2	6.0	5.9 ± 0.3	19.5 ± 0.4	3.3 ± 0.2
6.8	8.9 ± 0.4	21.4 ± 0.2	2.4 ± 0.1	6.8	6.6 ± 0.4	22.4 ± 0.5	3.4 ± 0.2
7.5	10.4 ± 1.4	28.6 ± 0.5	2.8 ± 0.4	7.5	10.8 ± 1.9	24.6 ± 3.2	2.3 ± 0.5
8.0	13.2 ± 0.7	41.4 ± 0.9	3.1 ± 0.2	8.0	14.6 ± 1.1	29.3 ± 3.4	2.0 ± 0.3
8.5	12.0 ± 1.1	41.6 ± 0.9	3.5 ± 0.3	8.5	12.6 ± 1.4	26.9 ± 2.8	2.1 ± 0.3
9.0	14.1 ± 0.9	36.1 ± 0.6	2.5 ± 0.2	9.0	10.6 ± 1.0	44.7 ± 4.0	4.2 ± 0.5
**H80A**				**H91A**			
**pH**	***k*_on_ (** **μM^−1^ s^−1^)**	***k*_off_ (s^−1^)**	**K_D_ (μM)**	**pH**	***k*_on_ (μM^−1^ s^−1^)**	***k*_off_ (s^−1^)**	**K_D_ (μM)**
5.0	4.6 ± 0.4	52.7 ± 3.2	11.5 ± 1.2	5.0	*	*	*
5.5	7.9 ± 0.9	20.0 ± 3.7	2.5 ± 0.5	5.5	5.6 ± 0.3	24.1 ± 0.4	4.3 ± 0.2
6.0	6.1 ± 0.3	19.6 ± 0.3	3.2 ± 0.1	6.0	6.6 ± 0.2	20.0 ± 0.3	3.0 ± 0.1
6.8	7.1 ± 0.4	21.6 ± 0.4	3.1 ± 0.2	6.8	7.8 ± 0.3	23.4 ± 0.4	3.0 ± 0.1
7.5	13.6 ± 1.1	27.8 ± 2.3	2.0 ± 0.2	7.5	10.0 ± 0.5	25.9 ± 0.3	2.6 ± 0.1
8.0	13.2 ± 0.7	30.1 ± 3.3	2.3 ± 0.3	8.0	13.2 ± 0.5	39.0 ± 0.7	2.9 ± 0.1
8.5	10.2 ± 1.2	32.5 ± 2.7	3.2 ± 0.5	8.5	13.6 ± 0.6	34.9 ± 0.5	2.6 ± 0.1
9.0	12.0 ± 0.9	34.0 ± 2.3	2.8 ± 0.3	9.0	12.7 ± 0.5	36.4 ± 0.5	2.9 ± 0.1

Note: (*) protein was not stable at this condition.

## Data Availability

Data available upon request.

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
