# Peer review of "Folding and Binding Mechanisms of the SH2 Domain from Crkl"

_biomolecules, 2022, doi:10.3390/biom12081014_

Round 1
Reviewer 1 Report
The manuscript by Nardella et al. describes the folding and stability of the SH2 domain from the Crk1 signaling protein. The authors perform a series of fluorescence-based assay, including denaturant unfolding and refolding experiments and ligand binding experiments. The authors identify that the Crk1 SH2 domain forms a folding intermediate in the presence of stabilizing sulfate. Moreover, the authors find that the interaction of the SH2 domain with a peptide derived from the protein Paxillin was stabilized by electrostatics. Finally, His60 of SH2 in the peptide binding pocket is important for the interaction.
Overall, this is a relatively straightforward piece of work that describes features of the folding and ligang binding of the Crk1 SH2 domain. The experiments were done rigorously, and the data are solid. I only have minor comments, presented below.
Minor comments:
1. Please provide information about the his-tag used for purification. Was it fused to the N- or C-terminal and was it cleaved off prior to experiments? If the his-tag remained on the protein, does its presence affect the binding experiments?
2. It appears that the Paxillin peptide used had an intact N- and C-terminus that were not chemically capped. This likely induces an electrostatic dipole on the peptide that is not present in the full-length protein. Can the authors comment on how this might influence their results?
3. It would be useful to include in the legends of figures the number of replicates performed for each experiment. It is included in the methods but including it in the legends is also helpful.
4. Can the authors comment on how these studies might be helpful in understanding the biology? The authors provide clear rationale for why the SH2 domain is important for signaling and indicate that these studies would be important for understanding dysfunction. However, how theses studies inform on the biology and potential inhibitor design is lacking. It would be very helpful to elaborate specifically on how the results could be used for understanding the biology.
Author Response
The manuscript by Nardella et al. describes the folding and stability of the SH2 domain from the Crk1 signaling protein. The authors perform a series of fluorescence-based assay, including denaturant unfolding and refolding experiments and ligand binding experiments. The authors identify that the Crk1 SH2 domain forms a folding intermediate in the presence of stabilizing sulfate. Moreover, the authors find that the interaction of the SH2 domain with a peptide derived from the protein Paxillin was stabilized by electrostatics. Finally, His60 of SH2 in the peptide binding pocket is important for the interaction.
Overall, this is a relatively straightforward piece of work that describes features of the folding and ligang binding of the Crk1 SH2 domain. The experiments were done rigorously, and the data are solid. I only have minor comments, presented below.
We sincerely thank the reviewer for the positive comments
Minor comments:
- Please provide information about the his-tag used for purification. Was it fused to the N- or C-terminal and was it cleaved off prior to experiments? If the his-tag remained on the protein, does its presence affect the binding experiments?
We have now added the information requested. The His-tag is covalently bound to the N-terminus of the protein. Previous experience on other SH2 domains has shown that the presence/absence of the tag does not affect binding in a detectable manner; hence (also to increase protein solubility) we favour to maintain the tag. We have now added this information in the Materials and Methods section.
- It appears that the Paxillin peptide used had an intact N- and C-terminus that were not chemically capped. This likely induces an electrostatic dipole on the peptide that is not present in the full-length protein. Can the authors comment on how this might influence their results?
The Reviewer is perfectly right that minor changes could be induced by having a free N- and C-termini. On the other hand, for consistency with previously characterized SH2 domains, we chose to use this type of peptide.
In this context, we wish to notice that the pH dependence reported in Figure 5, which are critical for the analysis of our data, are highly unlikely to be affected by the free N- and C-termini, which typically display respectively a much higher and much lower pKa than that observed in our experiments. We have now clarified this point at page 7.
- It would be useful to include in the legends of figures the number of replicates performed for each experiment. It is included in the methods but including it in the legends is also helpful.
We added the info. Thanks for the suggestion.
- Can the authors comment on how these studies might be helpful in understanding the biology? The authors provide clear rationale for why the SH2 domain is important for signaling and indicate that these studies would be important for understanding dysfunction. However, how theses studies inform on the biology and potential inhibitor design is lacking. It would be very helpful to elaborate specifically on how the results could be used for understanding the biology.
Thanks for the suggestion. We have now added this information in the discussion section
Reviewer 2 Report
The work "Folding and binding mechanisms of the SH2 domain from Crkl" is an interesting report on the folding pathway of the SH2 domain from Crkl and its impact on binding. The authors' observation of an intermediate binding state to ionic strength is particularly interesting. Beyond this the work is generally well written and the work clearly described.
I do have a few minor complaints:
(1) the authors do not report on the quality of their curve fittings. The authors simply note that the curves fit well, but quality metrics for these fits are missing. It would be helpful to have these denoted in the main text.
(2) The authors' observation that with increasing ionic strength the K_on for binding is altered but the k_off is not is quite interesting and the authors' conclusion that electrostatics may be playing a key-role in the stabilization of binding intermediate states; however, this appears to not yet be fully explored. It would be helpful to bring to bear some degree of molecular modeling to assist in quantifying what the intermediate state may be and how the electrostatics are being modified, but this may be beyond the scopre of the current work.
Overall, I find the work interesting and suitable for publication, with the minor request that the quality of the fits be quantified and reported.
Author Response
The work "Folding and binding mechanisms of the SH2 domain from Crkl" is an interesting report on the folding pathway of the SH2 domain from Crkl and its impact on binding. The authors' observation of an intermediate binding state to ionic strength is particularly interesting. Beyond this the work is generally well written and the work clearly described.
I do have a few minor complaints:
(1) the authors do not report on the quality of their curve fittings. The authors simply note that the curves fit well, but quality metrics for these fits are missing. It would be helpful to have these denoted in the main text.
We have added this info as requested.
(2) The authors' observation that with increasing ionic strength the K_on for binding is altered but the k_off is not is quite interesting and the authors' conclusion that electrostatics may be playing a key-role in the stabilization of binding intermediate states; however, this appears to not yet be fully explored. It would be helpful to bring to bear some degree of molecular modeling to assist in quantifying what the intermediate state may be and how the electrostatics are being modified, but this may be beyond the scopre of the current work.
We fully agree with the Reviewer that it would be of interest to provide some structural information regarding the effects of ionic strength on the observed rate constant. As noted by the Reviewer, this would be at this stage out of the scope of the present manuscript as it would imply extensive molecular modelling and/or molecular dynamics simulations. We are currently performing site directed mutagenesis to explore these phenomena, which will be possibly reported in a future publication.
Overall, I find the work interesting and suitable for publication, with the minor request that the quality of the fits be quantified and reported.
Reviewer 3 Report
The authors of the manuscript "Folding and binding mechanisms of the SH2 domain from Crkl" describe
the folding and unfolding process of the SH2 domain by following the fluorescence of intrinsic trp fluorescence using
steady-state and stopped-flow methods. In addition, the binding of a fluorescently labeled peptide was investigated.
The manuscript is well-written, and the experiments and data-analysis very well performed.
Some additional experiments could have been performed, like CD-spectroscopy, but nevertheless, the conclusion are solid.
The Materials and methods can be improved by providing more details.
Remarks
General: inconsistent unit use: 50 mM and 50mM, etc.
Equation 1: References to the origin of equation, or how to derive this equation are not provided.
mD-N and RT (units?) are not defined.
Equation2: what are Kiu etc.
2.4
Why was here 0.5 M NaCl used in the pH-dependence, not 0.15 M Na2SO4?
What is the 475 nm cutoff filter? Transmission above or below 475 nm?
128: Would be nice to show the sequence of dansyl-Pax112-123, to confirm the absence of trp, which would hamper the FRET measurements.
The equilibrium unfolding experiments were fitted using a 2-state model (and equation).
The unfolding-refolding kinetics clearly shows a (at least) 3-state process. Why were no experiments performed to check if the intermediate also populates under equilibrium conditions with e.g. CD-measurements?
What kind of intermediate do the authors propose, on- or off-pathway? Were aggregation effects observed?
Some better pictures of the SH2 domain would be nice. An obvious observation could be that His 60 is close to the phosphorylated tyrosine (number?), and thus can influence the binding of the peptide.
Author Response
Reviewer 3
The authors of the manuscript "Folding and binding mechanisms of the SH2 domain from Crkl" describe
the folding and unfolding process of the SH2 domain by following the fluorescence of intrinsic trp fluorescence using
steady-state and stopped-flow methods. In addition, the binding of a fluorescently labeled peptide was investigated.
The manuscript is well-written, and the experiments and data-analysis very well performed.
Some additional experiments could have been performed, like CD-spectroscopy, but nevertheless, the conclusion are solid.
The Materials and methods can be improved by providing more details.
We thank the Reviewer for appreciating our work. By following the guildelines of the journal, we wrote the Materials and Methods section in a relatively succinct manner to avoid repetitions with respect to previous work.
Remarks
General: inconsistent unit use: 50 mM and 50mM, etc.
Thanks for noticing. We fixed that.
Equation 1: References to the origin of equation, or how to derive this equation are not provided.
mD-N and RT (units?) are not defined.
Equation2: what are Kiu etc.
We fixed these points regarding the equations
Why was here 0.5 M NaCl used in the pH-dependence, not 0.15 M Na2SO4?
Whilst sodium sulphate is stabilizer of native states of proteins, it also promotes precipitation. Hence, to explore the effects of changes in pH (which could also promote aggregation), we favoured to carry out the experiments in NaCl.
What is the 475 nm cutoff filter? Transmission above or below 475 nm?
Transmission above 475 nm. We have now clarified this point.
128: Would be nice to show the sequence of dansyl-Pax112-123, to confirm the absence of trp, which would hamper the FRET measurements.
The sequence of the peptide is reported at page 6. No Trp is present.
The equilibrium unfolding experiments were fitted using a 2-state model (and equation).
The unfolding-refolding kinetics clearly shows a (at least) 3-state process. Why were no experiments performed to check if the intermediate also populates under equilibrium conditions with e.g. CD-measurements?
Since the intermediate is only marginally stable, it can be identified only via kinetic experiments. More to the point, the quantitative analysis of the kinetic data predicts the intermediate not to be populated at any equilibrium conditions and, therefore, the two-state nature of the equilibrium transition is in very good agreement with the kinetic data. We did not perform a CD analysis because, being the SH2 domain a predominantly beta protein and given that the denaturation occurs at very high concentration of denaturant (which jeopardizes a quantitative analysis), the signal to noise ratio is much worse than fluorescence.
What kind of intermediate do the authors propose, on- or off-pathway? Were aggregation effects observed?
These are very important point raised by the reviewer. Indeeed the folding rate constants did not display a dependence on protein concentration, indicating absence of transient aggregation effects (we have now clarified this point at page X).
Since only one rate constant could be observed, it is impossible at this stage to distinguish whether the folding intermediate is an on- or off-pathway species. We have now clarified this point at page 4.
Some better pictures of the SH2 domain would be nice. An obvious observation could be that His 60 is close to the phosphorylated tyrosine (number?), and thus can influence the binding of the peptide.
The reviewer is perfectly correct in noticing that the critical role of His 60 is also mirrored by its proximity to the ligand as highlighted in the structure of the complex reported in Figure 5B. We have stressed this point in the revision.